# Early Life Stage Folic Acid Deficiency Delays the Neurobehavioral Development and Cognitive Function of Rat Offspring by Hindering De Novo Telomere Synthesis

**DOI:** 10.3390/ijms23136948

**Published:** 2022-06-22

**Authors:** Dezheng Zhou, Zhenshu Li, Yue Sun, Jing Yan, Guowei Huang, Wen Li

**Affiliations:** 1Department of Nutrition and Food Science, School of Public Health, Tianjin Medical University, Tianjin 300070, China; dezhengzhou@163.com (D.Z.); lizhenshu@tmu.edu.cn (Z.L.); sunyuejy163@163.com (Y.S.); huangguowei@tmu.edu.cn (G.H.); 2Tianjin Key Laboratory of Environment, Nutrition and Public Health, Tianjin 300070, China; yanjing@tmu.edu.cn; 3Department of Social Medicine and Health Administration, School of Public Health, Tianjin Medical University, Tianjin 300070, China

**Keywords:** folic acid, pregnancy, telomere, neurobehavioral development, cognition

## Abstract

Early life stage folate status may influence neurodevelopment in offspring. The developmental origin of health and disease highlights the importance of the period of the first 1000 days (from conception to 2 years) of life. This study aimed to evaluate the effect of early life stage folic acid deficiency on de novo telomere synthesis, neurobehavioral development, and the cognitive function of offspring rats. The rats were divided into three diet treatment groups: folate-deficient, folate-normal, and folate-supplemented. They were fed the corresponding diet from 5 weeks of age to the end of the lactation period. After weaning, the offspring rats were still fed with the corresponding diet for up to 100 days. Neurobehavioral tests, folic acid and homocysteine (Hcy) levels, relative telomere length in brain tissue, and uracil incorporation in telomere in offspring were measured at different time points. The results showed that folic acid deficiency decreased the level of folic acid, increased the level of Hcy of brain tissue in offspring, increased the wrong incorporation of uracil into telomeres, and hindered de novo telomere synthesis. However, folic acid supplementation increased the level of folic acid, reduced the level of Hcy of brain tissue in offspring, reduced the wrong incorporation of uracil into telomeres, and protected de novo telomere synthesis of offspring, which was beneficial to the development of early sensory-motor function, spatial learning, and memory in adolescence and adulthood. In conclusion, early life stage folic acid deficiency had long-term inhibiting effects on neurodevelopment and cognitive function in offspring.

## 1. Introduction

It has been conclusively established that folic acid supplementation prior to and during early pregnancy (up to 12 weeks of gestation) can prevent neural tube defects (NTDs). In recent years, low folate status in early life has been linked to the risk of numerous adverse health conditions throughout life, from congenital disabilities to cardiovascular disease, cancer, and cognitive dysfunction in the elderly [1,2,3,4,5,6,7]. However, the impact of folic acid on neurodevelopment is complicated, other than NTDs. Although the survival/lifespan of offspring might not be seriously affected, prenatal folic acid had a profound effect on the neurodevelopment and cognition of offspring, but the mechanism remains elusive. Our previous study showed that maternal folate supplementation during pregnancy could stimulate neurobehavioral development in the infancy and adulthood of rat offspring [8]. However, the relationship and mechanisms of early life folic acid and offspring neurodevelopment were still unclear.

As a cofactor that mediates the transfer of one-carbon unit, folic acid plays a crucial role in nucleotide synthesis, DNA repair, and epigenetic modification [9,10]. Folic acid is involved in the biosynthesis of purines and thymine [11,12]. Deoxythymidine monophosphate (dTMP) is essential in DNA synthesis and normal cell division [13]. dTMP is derived from deoxyuridine monophosphate (dUMP) reductive methylation, in which 5, 10-methylene tetrahydrofolic acid acts as a cofactor to provide a single carbon group and thymidylate synthase (TS) acts as a catalyst [14,15]. The dUMP/dTMP ratio will increase abnormally when folic acid is deficient [7,16]. Excessive uracil can replace thymine residues with DNA polymerase and mistakenly incorporate them into DNA [16,17], leading to DNA strand breakage [7,17,18]. Studies have shown that DNA damage could lead to neuronal apoptosis [19].

Telomere, located at the end of chromosomes, is a nuclear protein complex formed by TTAGGG repeats and related proteins, which plays an essential role in preventing double-strand break and end-to-end fusion of chromosomes [19,20]. Folic acid levels are related to DNA synthesis, affecting telomere length [21]. The thymine-rich nature of telomere makes it more susceptible to uracil misincorporation, especially when folate is deficient [22]. This study hypothesized that early life folic acid deficiency obstructed de novo DNA synthesis, resulting in shorter telomere length in the offspring of infancy, and this effect might persist into adulthood, affecting neurobehavioral development.

Furthermore, in the real world, eating habits have continuity, which means offspring will have the same eating habits as their older generation. To simulate the natural state, we extended the intervention period of folic acid, further exploring the mechanism of early life folate nutritional status affecting offspring neurodevelopment. The present study hypothesized that early life folic acid deficiency hindered de novo telomere synthesis and delayed the rat offspring’s neurobehavioral development and cognitive function.

## 2. Results

### 2.1. Early Life Folic Acid Deficiency Decreased the Level of Folate and Increased the Level of Hcy in the Brain Tissue of Offspring

In postnatal day 0 (PND0), in PND14, PND60, and PND100 offspring, the levels of folate in brain tissue of the FA-N group and the FA-S group were higher than that of the FA-D group (all *p* < 0.05, Figure 1A–D); and compared with the FA-N group, folic acid supplementation increased the levels of folate in the brain tissue (all *p* < 0.05, Figure 1A–D).

In the offspring of PND0, the concentration of Hcy in the brain tissue of the FA-D group was significantly higher than that of the other two treatment groups (all *p* < 0.05, Figure 1E). At PND14, PND60, and PND100, the FA-D group had the highest concentration of Hcy in the brain tissue compared with the FA-N and FA-S groups (all *p* < 0.05, Figure 1F–H); and compared with the FA-N group, folic acid supplementation decreased the level of Hcy in the brain tissue (all *p* < 0.05, Figure 1F–H).

These results showed that early life folic acid deficiency decreased folate level and increased the Hcy level in the offspring’s brain tissue; however, folic acid supplementation increased folate level and decreased the Hcy level in the offspring’s brain tissue.

### 2.2. Early Life Folic Acid Deficiency Delayed the Neurodevelopment and Spatial Learning/Memory Abilities of the Offspring

Neurodevelopment of the childhood offspring was evaluated by righting reflexes on the surface and negative geotaxis test. The offspring PND4 to PND8 of the three different treatment groups showed progressive improvement in the righting reflex on the surface and negative geotaxis test. In the righting reflex on the surface, repeated-measures ANOVA showed that the reaction time of the offspring of the FA-N group and the FA-S group was shorter than that of the FA-D group (*p* < 0.05, Figure 2A), but there was no significant difference between the FA-N group and the FA-S group. In the negative geotaxis test, repeated-measures ANOVA also showed that the response time of the offspring of the FA-N group and the FA-S group was shorter than that of the FA-D group (*p* < 0.05, Figure 2B); moreover, the response time of the FA-S group was even shorter than the FA-N group (*p* < 0.05, Figure 2B).

The Morris water maze (MWM) test was used to assess the spatial learning and memory abilities of adolescent (PND50) and adult (PND90) offspring. During the phase of the acquisition test, both offspring PND50 and PND90 showed that the escape latency gradually shortened with the increase of experimental days. Through repeated-measures ANOVA, we found that in the offspring PND50 and PND90, the FA-D group had the most prolonged escape latency compared with the FA-N and FA-S groups (*p* < 0.05, Figure 3A,D); compared with the FA-S group, the FA-N group took longer time to the platform (*p* < 0.05, Figure 3A,D). In the spatial probe trial phase, we found that in the offspring PND50 and PND90, the FA-S group stayed longer in the target quadrant compared with the FA-D group (*p* < 0.05, Figure 3B,E) and more times to cross the platform (*p* < 0.05, Figure 3C,F); compared with the FA-D group, the FA-N group crossed the platform more times (*p* < 0.05, Figure 3C,F) and there was no significant difference between FA-N group and FA-S group.

In conclusion, neurobehavioral experiments showed that early life folic acid deficiency effectively delayed offspring’s sensory-motor function and spatial learning and memory abilities; however, folic acid supplementation improved the offspring’s neurodevelopment and spatial learning/memory abilities.

### 2.3. Early Life Folic Acid Deficiency Hindered De Novo Telomere Synthesis in the Brain of Offspring

The quantitative real-time polymerase chain reaction (qPCR) results showed that in the offspring of PND0, PND14, PND60, and PND100, for different diet intervention groups, the telomere lengths of FA-N and FA-S groups were longer than those of the FA-D group (all *p* < 0.05, Figure 4); compared with FA-N group, FA-S group had less severe telomere attrition (all *p* < 0.05, Figure 4). The telomere lengths gradually decreased with the age of the rat.

The results showed that early life folic acid deficiency hindered de novo telomere synthesis; however, folic acid supplementation alleviated telomere wear in brain tissue of rat offspring.

### 2.4. Early Life Folic Acid Deficiency Increased Uracil Misincorporation in Telomeres in Brain Tissue of Offspring

The levels of dUMP in brain tissue progressively increased with age. In the offspring of PND0, PND14, PND60, and PND100, compared with the FA-D group, folic acid supplementation reduced the levels of dUMP in brain tissue (all *p* < 0.05, Figure 5A); in the PND100 offspring, the dUMP level in the brain tissue of the FA-D group was also higher than that of the FA-N group (*p* < 0.05, Figure 5A).

At PND14 and PND100, compared with the FA-D group, folic acid supplementation increased the levels of dTMP in the brain tissue of offspring (all *p* < 0.05, Figure 5B); whereas in PND100 offspring, the level of dTMP in brain tissue of FA-D group was also lower than the FA-N group (*p* < 0.05, Figure 5B). In the offspring of PND14, PND60, and PND100, the dUMP/dTMP ratios of the FA-N group and FA-S group were lower than that of the FA-D group (all *p* < 0.05, Figure 5C); compared with the FA-N group, the dUMP/dTMP ratios of FA-S group were decreased (all *p* < 0.05, Figure 5C). At PND0, the dUMP/dTMP ratios of the FA-N group and FA-S group were lower than those of the FA-D group (*p* < 0.05, Figure 5C).

The relative uracil levels in the telomeres of the FA-D group were higher than those of the FA-N and FA-S groups at PND0, PND14, and PND60 (all *p* < 0.05, Figure 5D). In the PND100 offspring, the relative uracil levels in the telomeres of the three different treatment groups were significantly different, and the FA-D group had the highest level (*p* < 0.05, Figure 5D). The dUMP and dUMP/dTMP ratio increased with offspring age, which was alleviated by folic acid supplementation. The relative uracil levels in telomeres increased with the age of the offspring, and the supplementation of folic acid can reduce the uracil misincorporation in telomeres.

These results indicated that early life folic acid deficiency increased the level of dUMP, decreased dTMP in the brain tissue of the offspring, increased the dUMP/dTMP ratio, and increased the erroneous incorporation of uracil in the telomeres of the offspring’s brain tissue. However, folic acid supplementation improved the increase of the dUMP/dTMP ratio and reduced the erroneous incorporation of uracil in the telomeres of the offspring’s brain tissue which might be a potential mechanism for alleviating telomere attrition.

### 2.5. Early Life Folic Acid Deficiency Inhibited the Expression of Thymidylate Synthase in the Brain Tissue of Offspring

In the offspring of PND0, PND14, PND60, and PND100, the relative protein expressions of TS in the FA-S group were higher than that in the FA-D group (all *p* < 0.05, Figure 6A,B). Moreover, in PND14 offspring, the relative protein expression of TS in the FA-S group was also higher than that in the FA-N group (*p* < 0.05, Figure 6A,B).

The results showed that early life folic acid deficiency inhibited the expression of TS in the brain tissue of offspring; however, folic acid supplementation increased the expression of TS.

## 3. Discussion

This study showed that early life folic acid deficiency inhibited rat offspring’s neurobehavioral development and cognitive function. The potential hindering mechanism was de novo telomere synthesis. Early life folic acid deficiency lowered folate level and increased Hcy level in brain tissue, decreased the expression of TS, increased the accumulation of dUMP, decreased the concentration of dTMP, abnormally increased dUMP/dTMP ratio, aggravated the mismatch of uracil in brain telomeres, and hindered de novo telomere synthesis of rat offspring. However, folic acid supplementation increased the expression of TS, reduced the accumulation of dUMP, improved the abnormal increase of dUMP/dTMP ratio, reduced the mismatch of uracil in brain telomeres of rat offspring, and was in favor of de novo telomere synthesis, therefore accelerating neurobehavioral development and cognitive function of rat offspring in childhood, adolescence, and adulthood.

Studies showed that folic acid played an indispensable role in DNA synthesis and methylation, critical processes in neurodevelopment [23]. Moreover, maternal folate level is closely related to the neurobehavioral development of offspring [10,23,24]. The optimal folate intake of the mother during pregnancy was positively correlated with the offspring’s cognition function and motor development, whether in humans or animals [23]. Studies showed that inadequate folic acid intake during pregnancy increased the risk of NTDs and other neurodevelopmental disorders such as autism spectrum disorder (ASD) [25,26,27]. In addition, blood Hcy level was a risk factor for cognitive decline [28], and folic acid supplementation could improve memory [29]. This study showed that early life folic acid deficiency inhibited the rat offspring’s neurobehavioral development and cognitive function, consistent with previous studies [8].

Folate acts as a carbon donor and plays a reducing role [30], and TS plays a catalytic role in dUMP methylation to dTMP [14,15,30], which is essential for DNA synthesis and repair [13,22,31]. This process is the only de novo source of thymidine and the rate-limiting step in DNA synthesis [32,33]. Studies found that folic acid deficiency could lead to insufficient methylation of dUMP to dTMP, resulting in an abnormal increase in the ratio of dUMP/dTMP, and resulting in uracil instead of thymine being incorrectly incorporated into DNA [16,18]. Excessive incorporation of uracil in DNA may lead to severe consequences, such as point mutations, single-strand, double-strand breaks, chromosome breakage, and micronucleus formation [17]. This study showed that early life folic acid deficiency decreased the expression of TS and dTMP levels in the brain tissue of rat offspring. Nevertheless, the supplementation of folic acid increased the expression of TS and the levels of dTMP in brain tissue. However, the mechanism of TS increase caused by folic acid supplementation was unclear and needed to be further explored.

Telomeres are the six base-pair TTAGGG repetitive DNA sequences that cover and protect the ends of eukaryotic chromosomes [34]. Telomere length will gradually shorten with cell division times [34,35]. Telomere shortening is a marker of cellular senescence [20,34]. Studies reported that longer telomere length was associated with better cognitive performance [36]. Folate may affect telomere length by affecting DNA integrity [21,37]. Telomeres are rich in thymine, making them more susceptible to the wrong combination of uracil, causing damage to DNA and telomeres [22]. The present study showed that the offspring of maternal folate-deficient rats had shorter telomere length since birth. Furthermore, age would make this phenomenon even more prominent. The potential mechanism was that early life, folic acid deficiency hindered de novo telomere synthesis, and folic acid supplementation might reduce the shortening of telomere length by reducing the mismatch of uracil in the telomeres of brain tissue.

Our previous in vivo studies had confirmed that compared with supplementation limited to the periconceptional period, prolonged maternal folic acid supplementation throughout pregnancy was more beneficial to the neurobehavioral development of offspring [8]. Low folate status in early life is linked to the risk of numerous adverse health conditions throughout life, from congenital disabilities to cardiovascular disease, cancer, and cognitive dysfunction in the elderly [1,2,3,4,5,6,7]. However, the impact of folic acid on neurodevelopment is complicated, other than NTDs. Although the survival/lifespan of offspring might not be seriously affected, prenatal folic acid had a profound effect on the neurodevelopment and cognition of offspring. In this study, to further explore the long-term effects of folic acid on the neurobehavioral development of offspring, the duration of folic acid intervention was not limited to the pregnancy period but ran throughout the two months before pregnancy, the entire pregnancy period, and after weaning to postnatal 100 days. The data of this study showed that early life folic acid deficiency inhibited neurobehavioral development and cognitive function of rat offspring in childhood, adolescence, and adulthood, resulting in a series of adverse consequences.

Our previous in vitro studies found that folic acid supplementation decreased apoptosis in astrocytes via alleviating telomere attrition and telomere DNA oxidation damage [38,39]. However, other than astrocytes, the brain tissue is complicated. The natural development of the human embryo is challenging to study and cannot be only studied by in vitro research. This study aimed to discuss the effect of early life folic acid on neurodevelopment and telomere synthesis through maternal and offspring diet. The object of this study was completely different from previous studies. Furthermore, this study maximally simulated the natural human diet and investigated the effect of in vivo folate levels on early brain development in animals rather than on individual cells. The present study demonstrated the overall adverse effect of maternal or offspring early life folic acid deficiency in the offspring’s brain.

In conclusion, early life folic acid deficiency inhibited rat offspring’s neurobehavioral development and cognitive function; the potential mechanism was hindering de novo telomere synthesis. Folic acid supplementation improved rat offspring’s neurobehavioral development. Therefore, we suggest that women of childbearing age should pay attention to the nutritional status of folate before and during pregnancy. An appropriate amount of folate should also be taken by infants and toddlers, which may have a long-term beneficial impact on neurodevelopment and cognitive function.

## 4. Materials and Methods

### 4.1. Animals and Dietary Treatment

The Tianjin Medical University Animal Ethics Committee approved the experimental protocols of this study (TMUaEC2017003). Five-week-old female rats were randomly divided into three treatment groups (10 rats/group): (1) Folate-deficient diet group (FA-D) fed the folate-deficient diet; (2) Folate-normal diet group (FA-N) fed the folate-normal diet; and (3) Folate-supplemented diet group (FA-S) fed the folate-supplemented diet. Dietary treatment began at five weeks of age. After two months of dietary treatment, the dams were mated with male rats with a 2:1 female-to-male ratio. After the delivery of their pups, all dams were fed the corresponding diet (Figure 7A). Some of the offspring were fed the corresponding diet for up to 100 days (Figure 7B). In this study, the duration of folic acid intervention was not limited to the pregnancy period but ran throughout the two months before pregnancy, the entire pregnancy period, and after weaning to postnatal 100 days. All rats were housed in a specific pathogen-free facility with a temperature of (24 ± 2) °C, the humidity of (50 ± 5)%, and a light/dark cycle of 12 h, allowing ad libitum access to food and water until sacrifice. The offspring were euthanized with CO_2_ and sacrificed at PND0, PND14, PND60, and PND100. Their blood was collected in coagulant tubes and centrifuged to obtain serum and then stored at −80 °C; brain tissue was quickly frozen in liquid nitrogen and stored at −80 °C for subsequent biochemical tests.

All three diets were obtained from the Trophic Animal Feed High-Technology Company (Nantong, China). A folate-deficient diet, folate-normal diet, and folate-supplemented diet contain folic acid < 0.1, 2.0, and 3.5 mg/kg diet, were administered, respectively. The folate-supplemented diet added 1.5 mg folic acid/kg compared to the folate-normal diet, which in rats is equivalent to consuming a 400 μg folic acid supplement daily for a healthy diet in humans [40].

### 4.2. Folate and Homocysteine (Hcy) Assay

According to the manufacturer’s instructions, the folate level in brain tissue was measured by a competitive protein binding assay using an automated chemiluminescence system (Siemens immune 2000 xpi, Berlin, Germany). The system can detect all types of folate with a detection sensitivity limit of 0.8 ng/mL. The concentration of Hcy in brain tissue was assayed using Auto-Chemistry Analyzer (CS-T300, DIRUI, Changchun, China) with a detection sensitivity limit of 0.33 µmol/L. The isolated brain tissue was crushed with liquid nitrogen, and the homogenate was centrifuged (1:10, *w*/*v*, diluted with ice-cold normal saline) to obtain the supernatant, which was used for the determination of folate and Hcy. Protein quantification was performed using bicinchoninic acid (BCA) protein quantitative Kit (Shandong Sparkjade Biotechnology Co., Ltd., Dongying, China), and then the measured folate concentration was normalized to the measured protein concentration.

### 4.3. Neurobehavioral Tests

#### 4.3.1. Righting Reflex on the Surface

Righting reflex on the surface was performed on offspring rats on PND4, PND5, PND6, PND7, and PND8 [41]. This experiment can evaluate the sensory-motor function of offspring. The rats were turned over, and the spinal position was in contact with the plane. The time when the rats turned to the normal prone position and all four paws made contact with the plane was recorded. The test was terminated until the rats were not suitable for more than 30 s, and the time was recorded as 30 s.

#### 4.3.2. Negative Geotaxis Test

The negative geotaxis test was detected on offspring rats on PND4, PND5, PND6, PND7, and PND8 to assess sensory-motor function as the orienting response and movement expressed in opposition to clues of the gravitational vector [42]. The rats’ heads were placed down on a 30-degree inclined plane and the time it took for them to complete a 180-degree turn, the head-up position was recorded. If the rats did not complete the turn within 60 s, the test was considered negative and recorded for 60 s.

#### 4.3.3. Morris Water Maze Test

The MWM test [43] was used to test the spatial learning and memory ability of offspring rats at PND50 and PND90. This study used the WMT-100S Morris water maze test system (Chengdu Techman Software Co., Ltd., Chengdu, China). The test is divided into two phases: an acquisition test and a spatial probe trial. The system has a tank diameter of 1.2 m, averagely divided into four quadrants, and the platform was placed in the southeast quadrant. The acquisition test was conducted for five consecutive days. Rats were released from the designated position in each quadrant every day, and the distance and time when the rats swam and climbed onto the platform within 90 s were recorded, namely the escape latency. The rat, which could not reach the platform within 90 s, was placed on the platform for 15 s, then recorded as 90 s. The daily escape latency was the average of 4 quadrants. The spatial probe trial only lasted for one day. After the platform was removed, the rats were released from the northwest area of the tank and the number of times they passed the previous platform location and the time they spent in the platform location quadrant in 30 s was recorded.

### 4.4. Quantitative Real-Time Polymerase Chain Reaction (qPCR)

#### 4.4.1. Relative Telomere Length Detected by Quantitative Real-Time Polymerase Chain Reaction (qPCR)

The total DNA of brain tissue was extracted using a DNA extraction kit (Shandong Sparkjade Biotechnology Co., Ltd., Dongying, China) and quantified with a NanoDrop 2000 instrument (Thermo Fisher Scientific, Waltham, MA, USA), and all DNA samples to a concentration of 5 ng/μL were diluted. Four samples were randomly selected from each group, and 12 samples were finally selected to form a mixed genomic DNA to generate a standard curve. The single-copy gene AT1 was used to control each sample amplification [44,45]. The reaction used a 10 μL mixed system: DNA (1 μL, 5 ng/μL), 1 ×SYBR Green master mix (Promega Corporation, Madison, WI, USA) (5 μL), forward primer (0.4 μL, 10 μM), reverse primer (0.4 μL, 10 μM), and RNase-free water (3.2 μL). The reaction conditions were as follows: the mixture was first incubated at 95 °C for 10 min, and then 35 cycles were carried out (95 °C for 15 s and 60 °C for 1 min). Both telomere and AT1 used their specific primers [44,45] (GenScript, Nanjing, China). Telomere: forward, 5′-ggtttttgagggtgagggtgagggtgagggtgaggg-3′; reverse, 5′-tcccgactatccctatccctatccctatccctatcccta-3′. AT1: forward, 5′-acgtgttctcagcatcgaccgctacc-3′; reverse, 5′-agaatgataaggaaagggaacaagaagccc-3′. All samples were tested in the LightCycler 480 II instrument (Roche Applied Science, Basel, Switzerland).

#### 4.4.2. Uracil Misincorporation in Telomeres Detected by qPCR

DNA was isolated and quantified as described above. Uracil-DHA glycosylase (UDG) can recognize and excise uracil from telomeres [46]. Genomic DNA was incubated at 44 °C for 60 min in a buffer containing 20 mM Tris-HCl, 1 mM EDTA, and 1 mM DTT (pH 8.0), with or without UDG (1 U per 100 ng DNA). The reaction was terminated after incubation at 95 °C for 10 min. Finally, all DNA samples were diluted to 5ng/μL. The reaction used a 10 μL mixed system: DNA (1 μL, 5 ng/μL), 1 ×SYBR Green master mix (Promega Corporation, Madison, WI, USA) (12.5 μL), forward primer (1 μL, 10 μM), reverse primer (1 μL, 10 μM), and RNase-free water (9.5 μL). The reaction conditions were as follows: the mixture was first incubated at 95 °C for 10 min, and then 40 cycles were carried out (95 °C for 15 s and 60 °C for 1 min). Telomeres used their specific primers [44,45] (GenScript, Nanjing, China): forward, 5′-ggtttttgagggtgagggtgagggtgagggtgaggg-3′; reverse, 5′-tcccgactatccctatccctatccctatccctatcccta-3′. Ct values of genomic DNA treated with UDG and without UDG were compared. All samples were tested in the LightCycler 480 II instrument (Roche Applied Science, Basel, Switzerland).

### 4.5. dUMP and dTMP in Brain Tissue Detected by High-Performance Liquid Chromatography (HPLC)

The content of dUMP and dTMP in brain tissue was measured by HPLC [47]. An ultrasound cell breaker was used to break up brain tissue, which was then weighed, and found to be precisely 15 milligrams. Subsequently, the brain tissue was homogenized in 300 μL 0.1 M cold perchloric acid and centrifuged at 20,000× *g* for 20 min at 4 °C. The supernatant (250 μL) was transferred to another clean centrifuge tube, 62.5 μL of sodium carbonate (0.5 mol/L) was added and centrifuged at 4 °C, 20,000× *g* for 10 min. A disposable needle filter (0.22 μm, Millipore, Burlington, MA, USA) was used to filter the supernatant and inject the filtrate into an HPLC system (Waters Corporation, Milford, MA, USA), containing a Hypersil GOLDTM aQ column (Thermo Fisher Scientific, Waltham, MA, USA). A total of 0.005 mol/L potassium dihydrogen phosphate and 0.5% methanol by volume (adjusted pH to 2.5 with phosphoric acid) was used as the mobile phase to elute dUMP and dTMP at a flow rate of 0.26 mL/min and a column temperature of 24 °C. Elution compounds were monitored at 260 nm using an ultraviolet detector connected to a Hewlett-Packard HP3394 integrator. The detector can automatically integrate the peak area. The elution peaks were identified using dUMP and dTMP standards (Sigma Aldrich, St. Louis, MO, USA).

To remove residual perchloric acid, the precipitate was centrifuged at 15,000× *g* for 5 min at 4 °C. Precipitated protein was dissolved in 8mol/L urea solution (urea dissolved in 0.1 mol/L Tris-HCl and adjusted pH to 8.5 with concentrated hydrochloric acid). The mixed solution was incubated at 42 °C for 20 min and centrifuged at 150,000× *g* for 5 min. The protein concentration was determined by bicinchoninic acid (BCA) protein quantitative Kit (Shandong Sparkjade Biotechnology Co., Ltd., Dongying, China), and then the measured dUMP and dTMP concentrations were normalized with the measured protein concentration.

### 4.6. Western Blotting

The expression of TS in brain tissue was determined by Western blotting [48]. RIPA lysate was added to lyse the brain tissue cells and centrifuged at 14,000× *g* for 5 min after being processed by an ultrasonic disintegrator. A total of 10μL supernatant was diluted 10 times with PBS, and used a bicinchoninic acid (BCA) protein quantitative kit (Shandong Sparkjade Biotechnology Co., Ltd., Dongying, China) for protein quantification. SDS-PAGE was added to the same amount of protein and heated at 100 °C for 5–10 min. The protein was loaded on a 12% PAGE gel, separated in an electric field, and transferred to a PVDF membrane. The membrane was blocked with BSA for 1 h at room temperature. Then, the primary antibody was incubated and placed on a shaker overnight at 4 °C. After washing with TBST, the secondary antibody was incubated for 1 h at room temperature. Finally, the chemiluminescence reagent ECL was added, and the band was exposed to a Bio-Spectrum Gel Imaging System (Bio-Rad, Hercules, CA, USA) for photographing. The optical density was analyzed by Image J software, and the intensity of each protein band was normalized to the respective β-actin band.

### 4.7. Statistical Analysis

According to the data distribution, the data were expressed as proportion, mean ± SD. or P50 (P25, P75). Repeated measurements were compared using the repeated-measures ANOVA. Two-way ANOVA performed comparisons among different groups of the variables for factorial design. Statistical software SPSS 24.0 (IBM Corp., Armonk, NY, USA) was used for data analysis to evaluate differences between groups, which were considered statistically significant at *p* < 0.05.

## Figures and Tables

**Figure 1 ijms-23-06948-f001:**
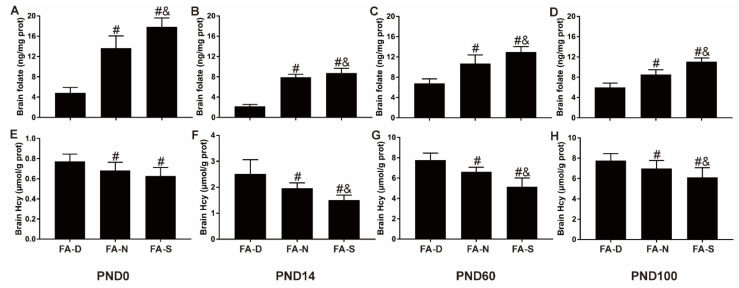
Folic acid supplementation increased the level of folate and decreased the level of Hcy in the brain tissue of offspring. Five-week-old female rats were randomly divided into three treatment groups (10 rats/group): (1) Folate-deficient diet group (FA-D) fed the folate-deficient diet; (2) Fo-late-normal diet group (FA-N) fed the folate-normal diet; (3) Folate-supplemented diet group (FA-S) fed the folate-supplemented diet. The duration of folic acid intervention was not limited to the pregnancy period but ran throughout the two months before pregnancy, the entire pregnancy period, and after weaning to postnatal 100 days. (**A**–**D**) The level of folate in brain tissue of offspring on PND0, PND14, PND60, and PND100. (**E**–**H**) The level of Hcy in brain tissue of offspring on PND0, PND14, PND60, and PND100. Data are expressed as mean ± SD (*n* = 10 brains/group). # *p* < 0.05 compared with FA-D group. & *p* < 0.05 compared with FA-N group.

**Figure 2 ijms-23-06948-f002:**
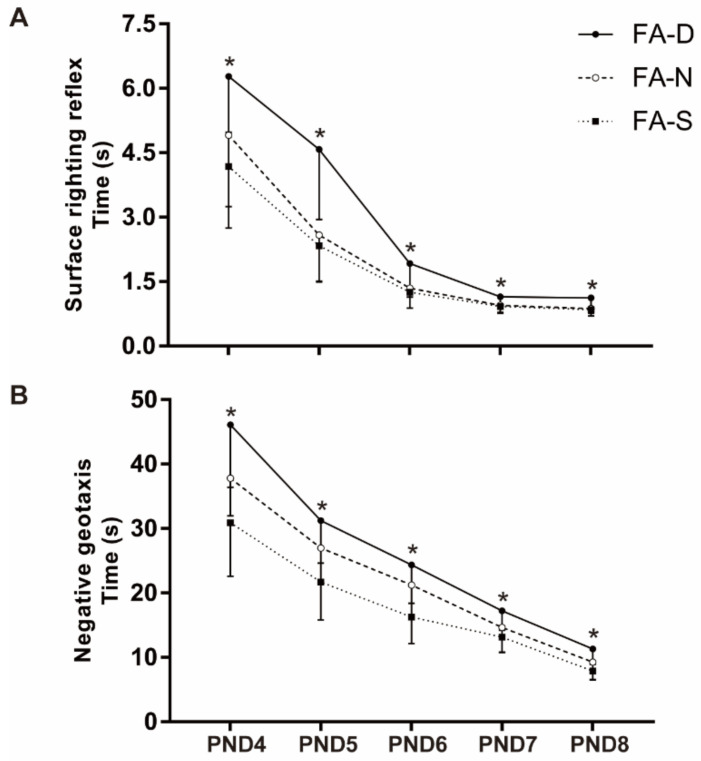
Folic acid supplementation improved sensory-motor function in infant offspring. Dams and pups were fed as described in Figure 1. Sixteen pups were selected randomly from each diet group (3–4 offspring from each dam) at PND4 and used for 5 consecutive days of righting reflex on the surface and negative geotaxis test. (**A**) Righting reflex on the surface for offspring. (**B**) Negative geotaxis test for offspring. Data are expressed as mean ± SD (*n* = 16 offspring for each group). * Comparing with three groups at α = 0.05/(comparison times).

**Figure 3 ijms-23-06948-f003:**
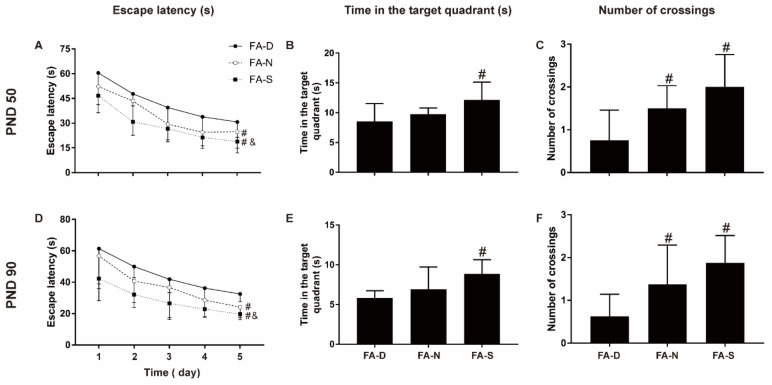
Folic acid improved spatial learning and memory ability in adolescent and adult offspring. Dams and pups were fed as described in Figure 1. (**A**) Escape latency during the spatial acquisition phase of Morris water maze (MWM) test on PND50. (**B**) Time in the targeted quadrant during spatial probe phase of MWM test on PND50. (**C**) Number of crossings during spatial probe phase on PND50. (**D**) Escape latency during the spatial acquisition phase of MWM test on PND90. (**E**) Time in the targeted quadrant during spatial probe phase of MWM test on PND90. (**F**) Number of crossings during spatial probe phase on PND90. Data are expressed as mean ± SD (*n* = 8 pups/group). # *p* < 0.05 compared with FA-D group. & *p* < 0.05 compared with FA-N group.

**Figure 4 ijms-23-06948-f004:**
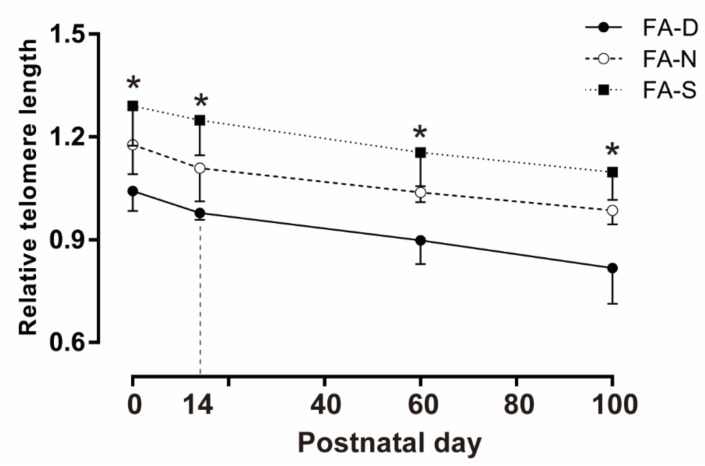
Folic acid supplementation alleviated telomere shortening in brain tissue of offspring. Dams and pups were fed as described in Figure 1. Brain tissue telomere length on PND0, PND14, PND60, and PND100 was quantified by qPCR. Data are expressed as mean ± SD (*n* = 8 brains/group). * Comparing with three groups at α = 0.05/(comparison times).

**Figure 5 ijms-23-06948-f005:**
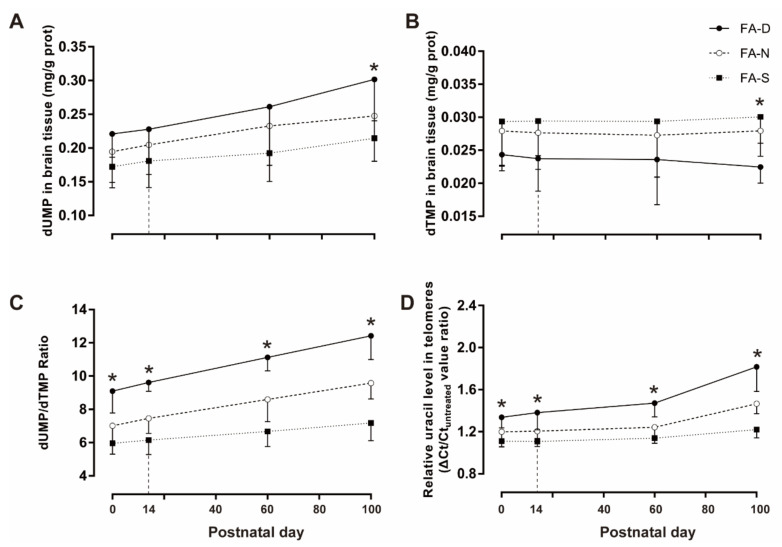
Folic acid supplementation reduced uracil misincorporation in telomeres in brain tissue of offspring. Dams and pups were fed as described in Figure 1. Brain dUMP level (**A**) and dTMP level (**B**) were determined by high-performance liquid chromatography, and dUMP/dTMP ratio (**C**) was calculated on PND0, PND14, PND60, and PND100. Relative uracil level in telomeres (**D**) was detected by quantitative PCR analysis on PND0, PND14, PND60, and PND100. Data are expressed as mean ± SD (*n* = 8 brains/group). * Comparing with three groups at α = 0.05/(comparison times).

**Figure 6 ijms-23-06948-f006:**
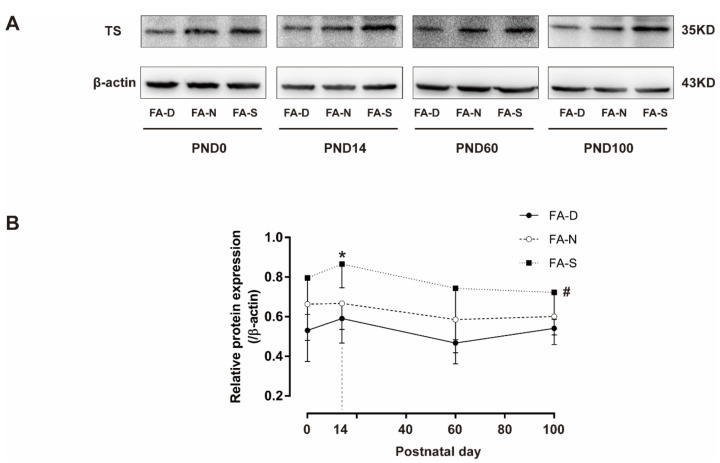
Folic acid supplementation increased the expression of TS in the brain tissue of offspring. Dams and pups were fed as described in Figure 1. (**A**) Representative western blotting of TS and β-actin on PND0, PND14, PND60, and PND100. (**B**) The protein expressions of TS quantified by western blotting were normalized to β-actin on PND0, PND14, PND60, and PND100. Data are expressed as mean ± SD (*n* = 5 brains/group). * *p* < 0.05, FA-S group compared with FA-N and FA-D group on PND14. # *p* < 0.05, FA-S group compared with FA-D group on PND0, PND14, PND60, and PND100.

**Figure 7 ijms-23-06948-f007:**
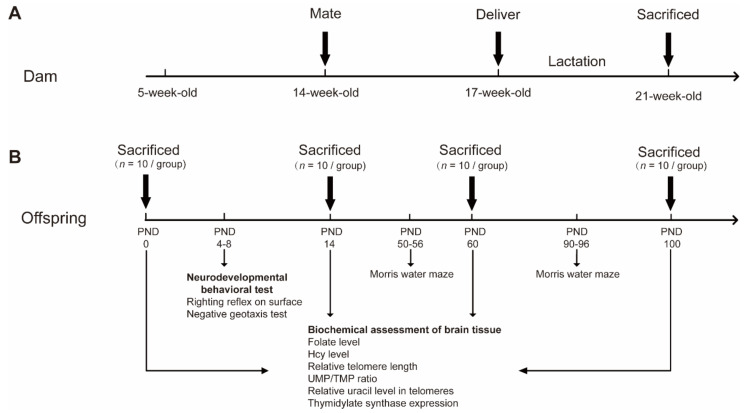
Flow chart of maternal and offspring experimental design. Three groups of the female rats were fed folate-normal, folate-deficient, or folate-supplemented diets before and during pregnancy, and the offspring were fed the corresponding diet after weaning. (**A**) Experimental design of dams. Dietary treatment began at 5 weeks of age. After two months of dietary treatment, the dams were mated with male rats with a 2:1 female-to-male ratio. After the delivery of their pups, all dams were fed the corresponding diet. After the lactation period, the dams were sacrificed. The duration of folic acid intervention was not limited to the pregnancy period but ran throughout the two months before pregnancy, the entire pregnancy period, and after weaning to postnatal 100 days. (**B**) Experimental design of offspring. The offspring were sacrificed and collected materials on PND0, PND14, PND60, and PND100. We performed the neurodevelopmental behavioral test on offspring rats on PND4, PND 5, PND 6, PND 7, and PND 8. At PND50 and PND90, we conducted a Morris water maze test on offspring rats.

## Data Availability

Not applicable.

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
