# Peer review of "Early Life Stage Folic Acid Deficiency Delays the Neurobehavioral Development and Cognitive Function of Rat Offspring by Hindering De Novo Telomere Synthesis"

_ijms, 2022, doi:10.3390/ijms23136948_

Round 1

Reviewer 1 Report

The manuscript by Zhou et al. has been improved. previously suggested experiments have not been included. the "original" blot pictures are not the raw data. please provide the raw images with marker.

Author Response

Thank you for your suggestion. Please see the attachment.

Reviewer 2 Report

This interesting manuscript describes how folic acid deficiency during pregnancy and in the early life stage may influence neurodevelopment in offspring using rats as animal model. Folic acid deficiency during pregnancy is well known to be involved in neural tube defects and, for this, folic acid is generally supplemented prior and during gestation to prevent such defects. However low folate state in early life has also been linked to the risk of several other adverse health conditions, among them cognitive dysfunctions in the elderly. The molecular mechanisms at the basis of folic acid deficiency on neurodevelopment and cognition are not well known. The authors show that delay in neurobehavioral development and cognitive functions may derive from the effect of folate deficiency on hindering de novo telomere synthesis.

The manuscript is well organized and results are clearly shown and explained and discussed.

I only suggest a throrough revision of English language prior to publication

Author Response

(The authors gave the same response as above.)

Round 2

Reviewer 1 Report

I agree that in the future the "merge" should be shown in the raw data. "in vivo" and "in vitro" should be in italics

This manuscript is a resubmission of an earlier submission. The following is a list of the peer review reports and author responses from that submission.

Round 1

Reviewer 1 Report

The manuscript by Zhou et al., presents a study on folate deficiency and neurodevelopment in rats. The study is designed with three different diets in dmas and offspring. The folate-deficiency is then analysed in regards to neurodevelopment and telemeres lenght.

The concept of folate-neurodevelopment relationship is widely known even if mechanism has not been fully dissected yet. There are a number of research papers on the subjects in mouse and chick models that should be listed in the references.

Although in vivo studies are of paramount importance when studying neurodevelopment, I would suggest to assess telomere lenght and/or DNA integrity also in vitro, considering the complicated network of microbiota and their folate production for corroborating results.

It would also be useful to assess plasma folate or HCY levels in diet-based studies.

Minor:

please add numbers of samples (brain/pups etc) used for each experiment.

if primers were designed ad hoc, please describe efficiency testing. If they were published elsewhere please cite relevant literature. Further, given the relevance of folate metabolites in DNA synthesis, please consider using 3 housekeeping genes for qPCR and geometric mean for normalization.

Author Response

Dear Reviewer 1,

Thank you for the opportunity to improve our manuscript entitled " Early life stage folic acid deficiency delays the neurobehavioral development and cognitive function of the rat offspring by hindering de novo telomere synthesis" (manuscript ID: ijms-1680496). We appreciate the valuable comments from you. We have carefully revised our manuscript according to those comments and have indicated the changes with red font in our manuscript. We provide below our detailed responses to your comments.

We are looking forward to hearing from you.

Best regards,

Sincerely,

Wen Li, Ph.D., MD

Professor

Department of Nutrition and Food Science

School of Public Health

Tianjin Medical University

22 Qixiangtai Road, Heping District,

Tianjin 300070, P.R. China

Tel: 86-22-83336603

Fax: 86-22-83336603

Response to Reviewer

Comments from the Reviewer 1:

Comment 1. The concept of folate-neurodevelopment relationship is widely known even if mechanism has not been fully dissected yet. There are a number of research papers on the subjects in mouse and chick models that should be listed in the references.

Response:

Thanks for your advice. Several references on the mouse model or chick model subjects have been cited in the reviewed manuscript.

References:

[12] Beaudin AE, Abarinov EV, Noden DM, Perry CA, Chu S, Stabler SP, Allen RH, Stover PJ. Shmt1 and de novo thymidylate biosynthesis underlie folate-responsive neural tube defects in mice. Am J Clin Nutr. 2011 Apr;93(4):789-98. doi: 10.3945/ajcn.110.002766.

[26] Wang T, Zhang T, Sun L, Li W, Zhang C, Yu L, Guan Y. Gestational B-vitamin supplementation alleviates PM2.5-induced autism-like behavior and hippocampal neurodevelopmental impairment in mice offspring. Ecotoxicol Environ Saf. 2019 Dec 15;185:109686. doi: 10.1016/j.ecoenv.2019.109686.

[27] Cai H, Lin L, Wang G, Berman Z, Yang X, Cheng X. Folic acid rescues corticosteroid-induced vertebral[Cai, 2020 #181] malformations in chick embryos through targeting TGF-β signaling. J Cell Physiol. 2020 Nov;235(11):8626-8639. doi: 10.1002/jcp.29707.

[36] Rolyan H, Scheffold A, Heinrich A, Begus-Nahrmann Y, Langkopf BH, Hölter SM, Vogt-Weisenhorn DM, Liss B, Wurst W, Lie DC, Thal DR, Biber K, Rudolph KL. Telomere shortening reduces Alzheimer's disease amyloid pathology in mice. Brain. 2011 Jul;134(Pt 7):2044-56. doi: 10.1093/brain/awr133.

Comment 2. Although in vivo studies are of paramount importance when studying neurodevelopment, I would suggest to assess telomere length and/or DNA integrity also in vitro, considering the complicated network of microbiota and their folate production for corroborating results.

Response:

Thanks for your advice. Our previous research involves the effects of folic acid on the telomere length and DNA integrity of astrocytes. Our previous studies have found that folic acid supplementation can alleviate the telomere attrition [1,2] and telomere DNA oxidation damage [2] of astrocytes.

The complicated network of microbiota and their folate production existed in rats, but the quantity of folate produced by intestinal flora is trace. And then the object of this study is to investigate the effect of early life stage folic acid deficiency on the neurobehavioral development and cognitive function of the rat offspring, so in vitro study was not used in this study.

References:

[1] Li Z, Zhou D, Zhang D, Zhao J, Li W, Sun Y, Chen Y, Liu H, Wilson JX, Qian Z, Huang G. Folic Acid Inhibits Aging-Induced Telomere Attrition and Apoptosis in Astrocytes In Vivo and In Vitro. Cereb Cortex. 2022 Jan 10;32(2):286-297. doi: 10.1093/cercor/bhab208.

[2] Li W, Ma Y, Li Z, Lv X, Wang X, Zhou D, Luo S, Wilson JX, Huang G. Folic Acid Decreases Astrocyte Apoptosis by Preventing Oxidative Stress-Induced Telomere Attrition. Int J Mol Sci. 2019 Dec 20;21(1):62. doi: 10.3390/ijms21010062.

Comment 3. It would also be useful to assess plasma folate or HCY levels in diet-based studies.

Response:

Thanks for your advice. Our previous study found serum folate concentration increased with folic acid supplementation in dams. Folic acid deficiency raised plasma Hcy concentration, but no significant differences were found in plasma Hcy concentration among the folate-supplemented and folate-normal diet groups. Moreover, maternal folic acid supplementation increased serum folate concentrations in pups, even the intervention only during pregnancy[1,2]. Furthermore, the concentration of folate and Hcy has consistency in brain tissue and serum[3]. So, the concentrations of folate and Hcy in plasma were not measured in this study.

References:

[1] Wang X, Li W, Li S, Yan J, Wilson JX, Huang G. Maternal Folic Acid Supplementation During Pregnancy Improves Neurobehavioral Development in Rat Offspring. Mol Neurobiol. 2018 Mar;55(3):2676-2684. doi: 10.1007/s12035-017-0534-2. Epub 2017 Apr 18. PMID: 28421540.

[2] Li W, Li Z, Li S, Wang X, Wilson JX, Huang G. Periconceptional Folic Acid Supplementation Benefit to Development of Early Sensory-Motor Function through Increase DNA Methylation in Rat Offspring. Nutrients. 2018 Mar 1;10(3):292. doi: 10.3390/nu10030292. PMID: 29494536; PMCID: PMC5872710.

[3] Lv X, Wang X, Wang Y, Zhou D, Li W, Wilson JX, Chang H, Huang G. Folic acid delays age-related cognitive decline in senescence-accelerated mouse prone 8: alleviating telomere attrition as a potential mechanism. Aging (Albany NY). 2019 Nov 22;11(22):10356-10373. doi: 10.18632/aging.102461. Epub 2019 Nov 22.

Comment 4. Please add numbers of samples (brain/pups etc) used for each experiment.

Response:

The sample size of each experiment have signed at line 89, line 139, line 153, line 190 and line 205.

Comment 5. If primers were designed ad hoc, please describe efficiency testing. If they were published elsewhere please cite relevant literature.

Response:

The primers [43,44] for telomere and AT1 were cited from references. And those works of literature were also cited in our manuscript (line 364, line 369 and line 385) and showed below.

References:

[43] Shoeb M, Kodali VK, Farris BY, Bishop LM, Meighan TG, Salmen R, Eye T, Friend S, Schwegler-Berry D, Roberts JR, Zeidler-Erdely PC, Erdely A, Antonini JM. Oxidative Stress, DNA Methylation, and Telomere Length Changes in Peripheral Blood Mononuclear Cells after Pulmonary Exposure to Metal-Rich Welding Nanoparticles. NanoImpact. 2017 Jan;5:61-69. doi: 10.1016/j.impact.2017.01.001.

[44] Antonini JM, Kodali V, Meighan TG, Roach KA, Roberts JR, Salmen R, Boyce GR, Zeidler-Erdely PC, Kashon M, Erdely A, Shoeb M. Effect of Age, High-Fat Diet, and Rat Strain on Serum Biomarkers and Telomere Length and Global DNA Methylation in Peripheral Blood Mononuclear Cells. Sci Rep. 2019 Feb 13;9(1):1996. doi: 10.1038/s41598-018-38192-0.

Comment 6. Given the relevance of folate metabolites in DNA synthesis, please consider using 3 housekeeping genes for c and geometric mean for normalization.

Response:

Thanks for your advice. We will consider using multiple housekeeping genes in our future research.

Reviewer 2 Report

This interesting manuscript describes how folic acid deficiency during pregnancy and in the early life stage may influence neurodevelopment in offspring using rats as animal model. Folic acid deficiency during pregnancy is well known to be involved in neural tube defects and, for this, folic acid is generally supplemented prior and during gestation to prevent such defects. However low folate state in early life has also been linked to the risk of several other adverse health conditions, among them cognitive dysfunctions in the elderly. The molecular mechanisms at the basis of folic acid deficiency on neurodevelopment and cognition are not well known. The authors show that delay in neurobehavioral development and cognitive functions may derive from the effect of folate deficiency on hindering de novo telomere synthesis.

The manuscript is well organized and results are clearly shown and explained and discussed. However I suggest some revisions prior to publication:

1- In the introduction, lines 67-76: I suggest to only explain the hypothesis of the study, while results should be written only in the Results section.

2- characters in fig 1 are too little, please increase the font

3- lines 149 and 150: Fig 4, not 4A

4-legend figure 4: “Brain tissue telomere length on PND0, PND14, 157 PND60, and PND100. Telomere length quantified by qPCR” please replace with “Brain tissue telomere length on PND0, PND14, 157 PND60, and PND100 was quantified by qPCR”

5- western blot figures are not very convincing, this is the most important question raised by this review. Also considering the original blot provided by authors, I am not so sure that TS is decreased in FA-D group. Please provide further experiments to strengthen this point.

6- Discussion: lines 254-257: “In the present study, early life folic acid deficiency hindered de novo telomere synthesis, but supplementation of folic acid may reduce the shortening of telomere length by reducing the mismatch of uracil in the telomeres of brain tissue” replace with “In the present study, we showed that etc…”

7- Discussion, lines 265 etc: I do not agree with this sentence “Even folate deficiency did not cause severe postnatal defects such as NTDs in the offspring”. Authors should consider a much higher number of rats to affirm this. It is well known that NTDs are not so frequent even in conditions of folate deficiency. Please discuss this point

8- Methods: please explain (as the authors explain in the discussion) that the duration of folic acid intervention was not limited to the pregnancy period but ran through the two months before pregnancy, the entire pregnancy period, and after weaning to postnatal 100 days. It is not clear in the text and in fig.7.

9- I suggest a deep revision of English language, due to several language and grammar errors or sentences not appropriately formulated, like in lines 54 and 63: “The dUMP/dTMP ratio will increase abnormally when folic acid deficiency.” or “Due to the specific structure of telomere riching in thymine, making it more susceptible cause damage by uracil misincorporation, especially when folate deficiency.” should be reformulated. Line 65, I suggest “…this effect may persist into adulthood…”. Line 162: I suggest: “…tissue progressively increases with age”. Several sentences in Methods are incomplete, like in line 379: “. Incubated genomic DNA at 44 ℃ for 60 min in a buffer containing 20 mM Tris-HCl, 1 mM EDTA, and 1 mM DTT (pH 380 8.0), with or without UDG (1 U per 100 ng DNA)” should be replaced with “ Genomic DNA was incubated …..”. Similarly for lines 397, 399, 400, 419, 420, 426.

10- line 427, I think filter not band

Round 2

Reviewer 1 Report

The revised manuscript has addressed some of the raised concern. However, despite some previous publications from the same group, the conclusions should be better corroborated by more data and/or refined methods, especially given the topic potentially of great interest.

Reviewer 2 Report

The manuscript can be accepted in the revised form